# Multi-Armed Bandits with Metric Movement Costs

**Tomer Koren**
Google Brain
tkoren@google.com

**Roi Livni**
Princeton University
rlivni@cs.princeton.edu

**Yishay Mansour**
Tel Aviv University and Google
mansour@cs.tau.ac.il

## Abstract

We consider the non-stochastic Multi-Armed Bandit problem in a setting where there is a fixed and known metric on the action space that determines a cost for switching between any pair of actions. The loss of the online learner has two components: the first is the usual loss of the selected actions, and the second is an additional loss due to switching between actions. Our main contribution gives a tight characterization of the expected minimax regret in this setting, in terms of a complexity measure $C$ of the underlying metric which depends on its covering numbers. In finite metric spaces with $k$ actions, we give an efficient algorithm that achieves regret of the form $\widetilde{O}(\max\{C^{1/3}T^{2/3}, \sqrt{kT}\})$, and show that this is the best possible. Our regret bound generalizes previous known regret bounds for some special cases: (i) the unit-switching cost regret $\widetilde{\Theta}(\max\{k^{1/3}T^{2/3}, \sqrt{kT}\})$ where $C = \Theta(k)$, and (ii) the interval metric with regret $\widetilde{\Theta}(\max\{T^{2/3}, \sqrt{kT}\})$ where $C = \Theta(1)$. For infinite metrics spaces with Lipschitz loss functions, we derive a tight regret bound of $\widetilde{\Theta}(T^{\frac{d+1}{d+2}})$ where $d \geq 1$ is the Minkowski dimension of the space, which is known to be tight even when there are no switching costs.

## 1 Introduction

Multi-Armed Bandit (MAB) is perhaps one of the most well studied model for learning that allows to incorporate settings with limited feedback. In its simplest form, MAB can be thought of as a game between a learner and an adversary: At first, the adversary chooses an arbitrary sequence of losses $\ell_1, \ldots, \ell_T$ (possibly adversarially). Then, at each round the learner chooses an action $i_t$ from a finite set of actions $K$. At the end of each round, the learner gets to observe her loss $\ell_t(i_t)$, and *only* the loss of her chosen action. The objective of the learner is to minimize her (external) regret, defined as the expected difference between her loss, $\sum_{t=1}^{T} \ell_t(i_t)$, and the loss of the best action in hindsight, i.e., $\min_{i \in K} \sum_{t=1}^{T} \ell_t(i)$.

One simplification of the MAB is that it assumes that the learner can switch between actions without any cost, this is in contrast to online algorithms that maintain a state and have a cost of switching between states. One simple intermediate solution is to add further costs to the learner that penalize *movements between actions*. (Since we compare the learner to the single best action, the adversary has no movement and hence no movement cost.) This approach has been studied in the MAB with unit switching costs [2, 12], where the learner is not only penalized for her loss but also pays a unit cost for any time she switches between actions. This simple penalty implicitly advocates the construction of algorithms that avoid frequent fluctuation in their decisions. Regulating switching has been successfully applied to many interesting instances such as buffering problems [16], limited-delay lossy coding [19] and dynamic pricing with patient buyers [15].

The unit switching cost assumes that any pair of actions have the same cost, which in many scenarios is far from true. For example, consider an ice-cream vendor on a beach, where his actions are to select a location and price. Clearly, changing location comes at a cost, while changing prices might come with no cost. In this case we can define a interval metric (the coast line) and the movement cost is the distance. A more involved case is a hot-dog vendor in Manhattan, which needs to select a location

and price. Again, it makes sense to charge a switching cost between locations according to their distance, and in this case the Manhattan-distance seems the most appropriate. Such settings are at the core of our model for MAB with movement cost. The authors of [24] considered a MAB problem equipped with an interval metric, i.e, the actions are $[0, 1]$ and the movement cost is the distance between the actions. They proposed a new online algorithm, called the Slowly Moving Bandit (SMB) algorithm, that achieves optimal regret bound for this setting, and applied it to a dynamic pricing problem with patient buyers to achieve a new tight regret bound.

The objective of this paper is to handle general metric spaces, both finite and infinite. We show how to generalize the SMB algorithm and its analysis to design optimal moving-cost algorithms for *any* metric space over finite decision space. Our main result identifies an intrinsic complexity measure of the metric space, which we call the *covering/packing complexity*, and give a tight characterization of the expected movement regret in terms of the complexity of the underlying metric. In particular, in finite metric spaces of complexity $C$ with $k$ actions, we give a regret bound of the form $\widetilde{O}(\max\{C^{1/3}T^{2/3}, \sqrt{kT}\})$ and present an efficient algorithm that achieves it. We also give a matching $\widetilde{\Omega}(\max\{C^{1/3}T^{2/3}, \sqrt{kT}\})$ lower bound that applies to *any* metric with complexity $C$.

We extend out results to general continuous metric spaces. For such a settings we clearly have to make some assumption about the losses, and we make the rather standard assumption that the losses are Lipchitz with respect to the underlying metric. In this setting our results depend on a quite different complexity measures: the upper and lower Minkowski dimensions of the space, thus exhibiting a phase transition between the finite case (that corresponds to Minkowski dimension zero) and the infinite case. Specifically, we give an upper bound on the regret of $\widetilde{O}(T^{\frac{d+1}{d+2}})$ where $d \geq 1$ is the *upper* Minkowski dimension. When the upper and lower Minkowski dimensions coincide—which is the case in many natural spaces, such as normed vector spaces—the latter bound matches a lower bound of [10] that holds even when there are no switching costs. Thus, a surprising implication of our result is that in infinite actions spaces (of bounded Minkowski dimension), adding movement costs do not add to the complexity of the MAB problem!

Our approach extends the techniques of [24] for the SMB algorithm, which was designed to optimize over an interval metric, which is equivalent to a complete binary Hierarchally well-Separated Tree (HST) metric space. By carefully balancing and regulating its sampling distributions, the SMB algorithm avoids switching between far-apart nodes in the tree and possibly incurring large movement costs with respect to the associated metric. We show that the SMB regret guarantees are much more general than just binary balanced trees, and give an analysis of the SMB algorithm when applied to general HSTs. As a second step, we show that a rich class of trees, on which the SMB algorithm can be applied, can be used to upper-bound any general metric. Finally, we reduce the case of an infinite metric space to the finite case via simple discretization, and show that this reduction gives rise to the Minkowski dimension as a natural complexity measure. All of these contractions turn out to be optimal (up to logarithmic factors), as demonstrated by our matching lower bounds.

## 1.1 Related Work

Perhaps the most well known classical algorithm for non-stochastic bandit is the EXP3 Algorithm [4] that guarantee a regret of $\widetilde{O}(\sqrt{kT})$ without movement costs. However, for general MAB algorithms there are no guarantees for slow movement between actions. In fact, it is known that in a worst case $\widetilde{\Omega}(T)$ switches between actions are expected (see [12]).

A simple case of MAB with movement cost is the uniform metric, i.e., when the distance between any two actions is the same. This setting has seen intensive study, both in terms of analyzing optimal regret rates [2, 12], as well as applications [16, 19, 15]. Our main technical tools for achieving lower bounds is through the lower bound of Dekel et al. [12] that achieve such bound for this special case. The general problem of bandits with movement costs has been first introduced in [24], where the authors gave an efficient algorithm for a 2-HST binary balanced tree metric, as well as for evenly spaced points on the interval. The main contribution of this paper is a generalization of these results to general metric spaces.

There is a vast and vigorous study of MAB in continuous spaces [23, 11, 5, 10, 32]. These works relate the change in the payoff to the change in the action. Specifically, there has been a vast research on Lipschitz MAB with stochastic payoffs [22, 29, 30, 21, 26], where, roughly, the expected reward is Lipschitz. For applying our results in continuous spaces we too need to assume Lipschitz losses,

however, our metric defines also the movement cost between actions and not only relates the losses of similar actions. Our general findings is that in Euclidean spaces, one can achieve the same regret bounds when movement cost is applied. Thus, the SMB algorithm can achieve the optimal regret rate.

One can model our problem as a deterministic Markov Decision Process (MDP), where the states are the MAB actions and in every state there is an action to move the MDP to a given state (which correspond to switching actions). The payoff would be the payoff of the MAB action associated with the state plus the movement cost to the next state. The work of Ortner [28] studies deterministic MDP where the payoffs are stochastic, and also allows for a fixed uniform switching cost. The work of Even-Dar et al. [13] and it extensions [27, 33] studies a MDP where the payoffs are adversarial but there is full information of the payoffs. Latter this work was extended to the bandit model by Neu et al. [27]. This line of works imposes various assumptions regarding the MDP and the benchmark policies, specifically, that the MDP is "mixing" and that the policies considered has full support stationary distributions, assumptions that clearly fail in our very specific setting.

Bayesian MAB, such as in the Gittins index (see [17]), assume that the payoffs are from some stochastic process. It is known that when there are switching costs then the existence of an optimal index policy is not guaranteed [6]. There have been some works on special cases with a fixed uniform switching cost [1, 3]. The most relevant work is that of Guha and Munagala [18] which for a general metric over the actions gives a constant approximation off-line algorithm. For a survey of switching costs in this context see [20].

The MAB problem with movement costs is related to the literature on online algorithms and the competitive analysis framework [8]. A prototypical online problem is the Metrical Task System (MTS) presented by Borodin et al. [9]. In a metrical task system there are a collection of states and a metric over the states. Similar to MAB, the online algorithm at each time step moves to a state, incurs a movement cost according to the metric, and suffers a loss that corresponds to that state. However, unlike MAB, in an MTS the online algorithm is given the loss prior to selecting the new state. Furthermore, competitive analysis has a much more stringent benchmark: the best sequence of actions in retrospect. Like most of the regret minimization literature, we use the best single action in hindsight as a benchmark, aiming for a vanishing average regret.

One of our main technical tools is an approximation from above of a metric via a Metric Tree (i.e., 2-HST). $k$-HST metrics have been vastly studied in the online algorithms starting with [7]. The main goal is to derive a simpler metric representation (using randomized trees) that will both upper and lower bound the given metric. The main result is to show a bound of $O(\log n)$ on the expected stretch of any edge, and this is also the best possible [14]. It is noteworthy that for bandit learning, and in contrast with these works, an upper bound over the metric suffices to achieve optimal regret rate. This is since in online learning we compete against the best *static* action in hindsight, which does not move at all and hence has zero movement cost. In contrast, in a MTS, where one compete against the best *dynamic* sequence of actions, one needs both an upper a lower bound on the metric.

## 2 Problem Setup and Background

In this section we recall the setting of Multi-armed Bandit with Movement Costs introduced in [24], and review the necessary background required to state our main results.

### 2.1 Multi-armed Bandits with Movement Costs

In the Multi-armed Bandits (MAB) with Movement Costs problem, we consider a game between an online learner and an adversary continuing for $T$ rounds. There is a set $K$, possibly infinite, of actions (or "arms") that the learner can choose from. The set of actions is equipped with a fixed and known metric $\Delta$ that determines a cost $\Delta(i, j) \in [0, 1]$ for moving between any pair of actions $i, j \in K$.

Before the game begins, an adversary fixes a sequence $\ell_1, \ldots, \ell_T : K \mapsto [0, 1]$ of loss functions assigning loss values in $[0, 1]$ to actions in $K$ (in particular, we assume an oblivious adversary). Then, on each round $t = 1, \ldots, T$, the learner picks an action $i_t \in K$, possibly at random. At the end of each round $t$, the learner gets to observe her loss (namely, $\ell_t(i_t)$) and nothing else. In contrast with the standard MAB setting, in addition to the loss $\ell_t(i_t)$ the learner suffers an additional cost due to her movement between actions, which is determined by the metric and is equal to $\Delta(i_t, i_{t-1})$. Thus, the total cost at round $t$ is given by $\ell_t(i_t) + \Delta(i_{t-1}, i_t)$.

The goal of the learner, over the course of $T$ rounds of the game, is to minimize her expected movement-regret, which is defined as the difference between her (expected) total costs and the total costs of the best fixed action in hindsight (that incurs no movement costs); namely, the *movement regret* with respect to a sequence $\ell_{1:T}$ of loss vectors and a metric $\Delta$ equals

$$\text{Regret}_{\text{MC}}(\ell_{1:T}, \Delta) = \mathbb{E}\left[\sum_{t=1}^{T} \ell_t(i_t) + \sum_{t=2}^{T} \Delta(i_t, i_{t-1})\right] - \min_{i \in K} \sum_{t=1}^{T} \ell_t(i).$$

Here, the expectation is taken with respect to the learner's randomization in choosing the actions $i_1, \ldots, i_T$; notice that, as we assume an oblivious adversary, the loss functions $\ell_t$ are deterministic and cannot depend on the learner's randomization.

## 2.2 Basic Definitions in Metric Spaces

We recall basic notions in metric space that govern the regret in the MAB with movement costs setting. Throughout we assume a bounded metric space $(K, \Delta)$, where for normalization we assume $\Delta(i, j) \in [0, 1]$ for all $i, j \in K$. Given a point $i \in K$ we will denote by $B_\epsilon(i) = \{j \in K : \Delta(i, j) \le \epsilon\}$ the ball of radius $\epsilon$ around $i$.

The following definitions are standard.

**Definition 1** (Packing numbers). A subset $P \subset K$ in a metric space $(K, \Delta)$ is an $\epsilon$-*packing* if the sets $\{B_\epsilon(i)\}_{i \in P}$ are disjoint sets. The $\epsilon$-*packing number* of $\Delta$, denoted $N_\epsilon^{\text{p}}(\Delta)$, is the maximum cardinality of any $\epsilon$-packing of $K$.

**Definition 2** (Covering numbers). A subset $C \subset K$ in a metric space $(K, \Delta)$ is an $\epsilon$-*covering* if $K \subseteq \cup_{i \in C} B_\epsilon(i)$. The $\epsilon$-*covering number* of $K$, denoted $N_\epsilon^{\text{c}}(\Delta)$, is the minimum cardinality of any $\epsilon$-covering of $K$.

**Tree metrics and HSTs.** We recall the notion of a tree metric, and in particular, a metric induced by an Hierarchically well-Separated (HST) Tree; see [7] for more details. Any weighted tree defines a metric over the vertices, by considering the shortest path between each two nodes. An HST tree (2-HST tree, to be precise) is a rooted weighted tree such that: 1) the edge weight from any node to each of its children is the same and 2) the edge weight along any path from the root to a leaf are decreasing by a factor 2 per edge. We will also assume that all leaves are of the same depth in the tree (this does not imply that the tree is complete).

Given a tree $\mathcal{T}$ we let $\text{depth}(\mathcal{T})$ denote its height, which is the maximal length of a path from any leaf to the root. Let $\text{level}(v)$ be the level of a node $v \in \mathcal{T}$, where the level of the leaves is 0 and the level of the root is $\text{depth}(\mathcal{T})$. Given nodes $u, v \in \mathcal{T}$, let $\text{LCA}(u, v)$ be their least common ancestor node in $\mathcal{T}$.

The metric which we next define is equivalent (up to a constant factor) to standard tree–metric induced over the leaves by an HST. By a slight abuse of terminology, we will call it HST metric:

**Definition 3** (HST metric). Let $K$ be a finite set and let $\mathcal{T}$ be a tree whose leaves are at the same depth and are indexed by elements of $K$. Then the HST metric $\Delta_{\mathcal{T}}$ over $K$ induced by the tree $\mathcal{T}$ is defined as follows:

$$\Delta_{\mathcal{T}}(i, j) = \frac{2^{\text{level}(\text{LCA}(i,j))}}{2^{\text{depth}(\mathcal{T})}} \qquad \forall\, i, j \in K.$$

For a HST metric $\Delta_{\mathcal{T}}$, observe that the packing number and covering number are simple to characterize: for all $0 \le h < \text{depth}(\mathcal{T})$ we have that for $\epsilon = 2^{h-H}$,

$$N_\epsilon^{\text{c}}(\Delta_{\mathcal{T}}) = N_\epsilon^{\text{p}}(\Delta_{\mathcal{T}}) = \left|\{v \in \mathcal{T} : \text{level}(v) = h\}\right|.$$

**Complexity measures for finite metric spaces.** We next define the two notions of complexity that, as we will later see, governs the complexity of MAB with metric movement costs.

**Definition 4** (covering complexity). The covering complexity of a metric space $(K, \Delta)$ denoted $C_{\text{c}}(\Delta)$ is given by

$$C_{\text{c}}(\Delta) = \sup_{0 < \epsilon < 1} \epsilon \cdot N_\epsilon^{\text{c}}(\Delta).$$

**Definition 5** (packing complexity). The packing complexity of a metric space $(K, \Delta)$ denoted $C_{\mathrm{p}}(\Delta)$ is given by

$$C_{\mathrm{p}}(\Delta) = \sup_{0 < \epsilon < 1} \epsilon \cdot N_\epsilon^{\mathrm{p}}(\Delta).$$

For a HST metric, the two complexity measures coincide as its packing and covering numbers are the same. Therefore, for a HST metric $\Delta_{\mathcal{T}}$ we will simply denote the complexity of $(K, \Delta_{\mathcal{T}})$ as $C(\mathcal{T})$. In fact, it is known that in any metric space $N_\epsilon^{\mathrm{p}}(\Delta) \leq N_\epsilon^{\mathrm{c}}(\Delta) \leq N_{\epsilon/2}^{\mathrm{p}}(\Delta)$ for all $\epsilon > 0$. Thus, for a general metric space we obtain that

$$C_{\mathrm{p}}(\Delta) \leq C_{\mathrm{c}}(\Delta) \leq 2C_{\mathrm{p}}(\Delta). \tag{1}$$

**Complexity measures for infinite metric spaces.** For infinite metric spaces, we require the following definition.

**Definition 6** (Minkowski dimensions). Let $(K, \Delta)$ be a bounded metric space. The upper Minkowski dimension of $(K, \Delta)$, denoted $\overline{\mathcal{D}}(\Delta)$, is defined as

$$\overline{\mathcal{D}}(\Delta) = \limsup_{\epsilon \to 0} \frac{\log N_\epsilon^{\mathrm{p}}(\Delta)}{\log(1/\epsilon)} = \limsup_{\epsilon \to 0} \frac{\log N_\epsilon^{\mathrm{c}}(\Delta)}{\log(1/\epsilon)}.$$

Similarly, the lower Minkowski dimension is denoted by $\underline{\mathcal{D}}(\Delta)$ and is defined as

$$\underline{\mathcal{D}}(\Delta) = \liminf_{\epsilon \to 0} \frac{\log N_\epsilon^{\mathrm{p}}(\Delta)}{\log(1/\epsilon)} = \liminf_{\epsilon \to 0} \frac{\log N_\epsilon^{\mathrm{c}}(\Delta)}{\log(1/\epsilon)}.$$

We refer to [31] for more background on the Minkowski dimensions and related notions in metric spaces theory.

## 3 Main Results

We now state the main results of the paper, which give a complete characterization of the expected regret in the MAB with movement costs problem.

### 3.1 Finite Metric Spaces

The following are the main results of the paper.

**Theorem 7** (Upper Bound). *Let $(K, \Delta)$ be a finite metric space over $|K| = k$ elements with diameter $\leq 1$ and covering complexity $C_c = C_c(\Delta)$. There exists an algorithm such that for any sequence of loss functions $\ell_1, \ldots, \ell_T$ guarantees that*

$$Regret_{\mathsf{MC}}(\ell_{1:T}, \Delta) = \widetilde{O}\left( \max\{C_c^{1/3} T^{2/3}, \sqrt{kT}\} \right).$$

**Theorem 8** (Lower Bound). *Let $(K, \Delta)$ be a finite metric space over $|K| = k$ elements with diameter $\geq 1$ and packing complexity $C_p = C_p(\Delta)$. For any algorithm there exists a sequence $\ell_1, \ldots, \ell_T$ of loss functions such that*

$$Regret_{\mathsf{MC}}(\ell_{1:T}, \Delta) = \widetilde{\Omega}\left( \max\{C_p^{1/3} T^{2/3}, \sqrt{kT}\} \right).$$

For the detailed proofs, see the full version of the paper [25]. Recalling Eq. (1), we see that the regret bounds obtained in Theorems 7 and 8 are matching up to logarithmic factors. Notice that the tightness is achieved *per instance*; namely, for any given metric we are able to fully characterize the regret's rate of growth as a function of the intrinsic properties of the metric. (In particular, this is substantially stronger than demonstrating a specific metric for which the upper bound cannot be improved.) Note that for the lower bound statement in Theorem 8 we require that the diameter of $K$ is bounded away from zero, where for simplicity we assume a constant bound of 1. Such an assumption is necessary to avoid degenerate metrics. Indeed, when the diameter is very small, the problem reduces to the standard MAB setting without any additional costs and we obtain a regret rate of $\Omega(\sqrt{kT})$.

Notice how the above results extend known instances of the problem from previous work: for uniform movement costs (i.e., unit switching costs) over $K = \{1, \ldots, k\}$ we have $C_{\mathrm{c}} = \Theta(k)$, so that the

obtain bound is $\widetilde{\Theta}(\max\{k^{1/3}T^{2/3}, \sqrt{kT}\})$, which recovers the results in [2, 12]; and for a 2-HST binary balanced tree with $k$ leaves, we have $C_c = \Theta(1)$ and the resulting bound is $\widetilde{\Theta}(\max\{T^{2/3}, \sqrt{kT}\})$, which is identical to the bound proved in [24].

The 2-HST regret bound in [24] was primarily used to obtain regret bounds for the action space $K = [0, 1]$. In the next section we show how this technique is extended for infinite metric space to obtain regret bounds that depend on the dimensionality of the action space.

## 3.2 Infinite Metric Spaces

When $(K, \Delta)$ is an infinite metric space, without additional constraints on the loss functions, the problem becomes ill-posed with a linear regret rate, even without movement costs. Therefore, one has to make additional assumptions on the loss functions in order to achieve sublinear regret. One natural assumption, which is common in previous work, is to assume that the loss functions $\ell_1, \ldots, \ell_T$ are all 1-Lipschitz with respect to the metric $\Delta$. Under this assumption, we have the following result.

**Theorem 9.** *Let $(K, \Delta)$ be a metric space with diameter $\leq 1$ and upper Minkowski dimension $d = \overline{\mathcal{D}}(\Delta)$, such that $d \geq 1$. There exists a strategy such that for any sequence of loss functions $\ell_1, \ldots, \ell_T$, which are all 1-Lipschitz with respect to $\Delta$, guarantees that*

$$Regret_{\mathsf{MC}}(\ell_{1:T}, \Delta) = \widetilde{O}\big(T^{\frac{d+1}{d+2}}\big).$$

We refer the full version of the paper [25] for a proof of the theorem. Again, we observe that the above result extend the case of $K = [0, 1]$ where $d = 1$. Indeed, for Lipschitz functions over the interval a tight regret bound of $\widetilde{\Theta}(T^{2/3})$ was achieved in [24], which is exactly the bound we obtain above.

We mention that a lower bound of $\widetilde{\Omega}(T^{\frac{d+1}{d+2}})$ is known for MAB in metric spaces with Lipschitz cost functions—even *without movement costs*—where $d = \underline{\mathcal{D}}(\Delta)$ is the lower Minkowski dimension.

**Theorem 10** (Bubeck et al. [10])**.** *Let $(K, \Delta)$ be a metric space with diameter $\leq 1$ and lower Minkowski dimension $d = \underline{\mathcal{D}}(\Delta)$, such that $d \geq 1$. Then for any learning algorithm, there exists a sequence of loss function $\ell_1, \ldots, \ell_T$, which are all 1-Lipschitz with respect to $\Delta$, such that the regret (without movement costs) is $\widetilde{\Omega}\big(T^{\frac{d+1}{d+2}}\big).$*

In many natural metric spaces in which the upper and lower Minkowski dimensions coincide (e.g., normed spaces), the bound of Theorem 9 is tight up to logarithmic factors in $T$. In particular, and quite surprisingly, we see that the movement costs do not add to the regret of the problem!

It is important to note that Theorem 9 holds only for metric spaces whose (upper) Minkowski dimension is at least 1. Indeed, finite metric spaces are of Minkowski dimension zero, and as we demonstrated in Section 3.1 above, a $O(\sqrt{T})$ regret bound is not achievable. Finite matric spaces are associated with a complexity measure which is very different from the Minkowski dimension (i.e., the covering/packing complexity). In other words, we exhibit a phase transition between dimension $d = 0$ and $d \geq 1$ in the rate of growth of the regret induced by the metric.

# 4 Algorithms

In this section we turn to prove Theorem 7. Our strategy is much inspired by the approach in [24], and we employ a two-step approach: First, we consider the case that the metric is a HST metric; we then turn to deal with general metrics, and show how to upper-bound any metric with a HST metric.

## 4.1 Tree Metrics: The Slowly-Moving Bandit Algorithm

In this section we analyze the simplest case of the problem, in which the metric $\Delta = \Delta_{\mathcal{T}}$ is induced by a HST tree $\mathcal{T}$ (whose leaves are associated with actions in $K$). In this case, our main tool is the Slowly-Moving Bandit (SMB) algorithm [24]: we demonstrate how it can be applied to general tree metrics, and analyze its performance in terms of intrinsic properties of the metric.

We begin by reviewing the SMB algorithm. In order to present the algorithm we require few additional notations. The algorithm receives as input a tree structure over the set of actions $K$, and its operation depends on the tree structure. We fix a HST tree $\mathcal{T}$ and let $H = \text{depth}(\mathcal{T})$. For any level $0 \leq h \leq H$ and action $i \in K$, let $A_h(i)$ be the set of leaves of $\mathcal{T}$ that share a common ancestor with $i$ at level $h$

(recall that level $h = 0$ is the bottom–most level corresponding to the singletons). In terms of the tree metric we have that $A_h(i) = \{j : \Delta_{\mathcal{T}}(i, j) \leq 2^{-H+h}\}$.

The SMB algorithm is presented in Algorithm 1. The algorithm is based on the multiplicative update method, in the spirit of EXP3 algorithms [4]. Similarly to EXP3, the algorithm computes at each round $t$ an estimator $\widetilde{\ell}_t$ to the loss vector $\ell_t$ using the single loss value $\ell_t(i_t)$ observed. In addition to being an (almost) unbiased estimate for the true loss vector, the estimator $\widetilde{\ell}_t$ used by SMB has the additional property of inducing slowly-changing sampling distributions $p_t$: This is done by choosing at random a level $h_t$ of the tree to be rebalanced (in terms of the weights maintained by the algorithm): As a result, the marginal probabilities $p_{t+1}(A_{h_t}(i))$ are not changed at round $t$.

In turn, and in contrast with EXP3, the algorithm choice of action at round $t + 1$ is not purely sampled from $p_t$, but rather conditioned on our last choice of level $h_t$. This is informally justified by the fact that $p_t$ and $p_{t+1}$ agree on the marginal distribution of $A_{h_t}(i_t)$, hence we can think of the level drawn at round $t$ as if it were drawn subject to $p_{t+1}(A_{h_t}) = p_t(A_{h_t})$.

---

Input: A tree $\mathcal{T}$ with a set of finite leaves $K$, $\eta > 0$.
Initialize: $H = \text{depth}(\mathcal{T})$, $A_h(i) = B_{2^{-H+h}}(i)$, $\forall i \in K, 0 \leq h \leq H$
Initialize $p_1 = \text{unif}(K)$, $h_0 = H$ and $i_0 \sim p_1$
For $t = 1, \ldots, T$:
    (1) Choose action $i_t \sim p_t(\cdot \mid A_{h_{t-1}}(i_{t-1}))$, observe loss $\ell_t(i_t)$
    (2) Choose $\sigma_{t,0}, \ldots, \sigma_{t,H-1} \in \{\pm 1\}$ uniformly at random;
        let $h_t = \min\{0 \leq h \leq H : \sigma_{t,h} < 0\}$ where $\sigma_{t,H} = -1$
    (3) Compute vectors $\bar{\ell}_{t,0}, \ldots, \bar{\ell}_{t,H-1}$ recursively via

$$\bar{\ell}_{t,0}(i) = \frac{\mathbb{1}\{i_t = i\}}{p_t(i)}\ell_t(i_t),$$

and for all $h \geq 1$:

$$\bar{\ell}_{t,h}(i) = -\frac{1}{\eta}\ln\left(\sum_{j \in A_h(i)}\frac{p_t(j)}{p_t(A_h(i))}e^{-\eta(1+\sigma_{t,h-1})\bar{\ell}_{t,h-1}(j)}\right)$$

    (4) Define $E_t = \{i : p_t(A_h(i)) < 2^h\eta$ for some $0 \leq h < H\}$ and set:

$$\widetilde{\ell}_t = \begin{cases} 0 & \text{if } i_t \in E_t; \\ \bar{\ell}_{t,0} + \sum_{h=0}^{H-1}\sigma_{t,h}\bar{\ell}_{t,h} & \text{otherwise} \end{cases}$$

    (5) Update:

$$p_{t+1}(i) = \frac{p_t(i)\,e^{-\eta\widetilde{\ell}_t(i)}}{\sum_{j=1}^{k}p_t(j)\,e^{-\eta\widetilde{\ell}_t(j)}} \qquad \forall\, i \in K$$

Algorithm 1: The SMB algorithm.

---

A key observation is that by directly applying SMB to the metric $\Delta_{\mathcal{T}}$, we can achieve the following regret bound:

**Theorem 11.** *Let $(K, \Delta_{\mathcal{T}})$ be a metric space defined by a 2-HST $\mathcal{T}$ with $\text{depth}(\mathcal{T}) = H$ and complexity $C(\mathcal{T}) = C$. Using SMB algorithm we can achieve the following regret bound:*

$$Regret_{\text{MC}}(\ell_{1:T}, \Delta_{\mathcal{T}}) = O\left(H\sqrt{2^H TC\log C} + H2^{-H}T\right). \tag{2}$$

To show Theorem 11, we adapt the analysis of [24] (that applies only to complete binary HSTs) to handle more general HSTs. We defer this part of our analysis to the full version of the paper [25], since it follows from a technical modification of the original proof.

For a tree that is either too deep or too shallow, Eq. (2) may not necessarily lead to a sublinear regret bound, let alone optimal. The main idea behind achieving optimal regret bound for a general tree, is to modify it until one of two things happen: Either we have optimized the depth so that the two terms in the left-hand side of Eq. (2) are of same order: In that case, we will show that one can achieve

regret rate of order $O(C(\mathcal{T})^{1/3}T^{2/3})$. If we fail to do that, we show that the first term in the left-hand side is the dominant one, and it will be of order $O(\sqrt{kT})$.

For trees that are in some sense "well behaved" we have the following Corollary of Theorem 11.

**Corollary 12.** *Let $(K, \Delta_{\mathcal{T}})$ be a metric space defined by a tree $\mathcal{T}$ over $|K| = k$ leaves with* depth$(\mathcal{T}) = H$ *and complexity* $C(\mathcal{T}) = C$. *Assume that $\mathcal{T}$ satisfies the following:*

*(1) $2^{-H}HT \leq \sqrt{2^H HCT}$;*
*(2) One of the following is true:*
    *(a) $2^H C \leq k$;*
    *(b) $2^{-(H-1)}(H-1)T \geq \sqrt{2^{H-1}(H-1)CT}$.*

*Then, the SMB algorithm can be used to attain* $Regret_{\mathsf{MC}}(\ell_{1:T}, \Delta_{\mathcal{T}}) = \widetilde{O}\big(\max\{C^{1/3}T^{2/3}, \sqrt{kT}\}\big)$.

The following establishes Theorem 7 for the special case of tree metrics.

**Lemma 13.** *For any tree $\mathcal{T}$ and time horizon $T$, there exists a tree $\mathcal{T}'$ (over the same set $K$ of $k$ leaves) that satisfies the conditions of Corollary 12, such that $\Delta_{\mathcal{T}'} \geq \Delta_{\mathcal{T}}$ and $C(\mathcal{T}') = C(\mathcal{T})$. Furthermore, $\mathcal{T}'$ can be constructed efficiently from $\mathcal{T}$ (i.e., in time polynomial in $|K|$ and $T$). Hence, applying SMB to the metric space $(K, \Delta_{\mathcal{T}'})$ leads to* $Regret_{\mathsf{MC}}(\ell_{1:T}, \Delta_{\mathcal{T}}) = \widetilde{O}\big(\max\{C(\mathcal{T})^{1/3}T^{2/3}, \sqrt{kT}\}\big)$.

We refer to [25] for the proofs of both results.

## 4.2 General Finite Metrics

Finally, we obtain the general finite case as a corollary of the following.

**Lemma 14.** *Let $(K, \Delta)$ be a finite metric space. There exists a tree metric $\Delta_{\mathcal{T}}$ over $K$ (with $|K| = k$) such that $4\Delta_{\mathcal{T}}$, dominates $\Delta$ (i.e., such that $4\Delta_{\mathcal{T}}(i, j) \geq \Delta(i, j)$ for all $i, j \in K$) for which $C(\mathcal{T}) = O(C_c(\Delta) \log k)$. Furthermore, $\mathcal{T}$ can be constructed efficiently.*

*Proof.* Let $H$ be such that the minimal distance in $\Delta$ is larger than $2^{-H}$. For each $r = 2^{-1}, 2^{-2}, \ldots, 2^{-H}$ we let $\{B_r(i_{\{1,r\}}), \ldots, B_r(i_{\{m_r, r\}})\} = \mathcal{B}_r$ be a covering of $K$ of size $N_r^{\mathsf{c}}(\mathcal{T}) \log k$ using balls of radius $r$. Note that finding a minimal set of balls of radius $r$ that covers $K$ is exactly the set cover problem. Hence, we can efficiently approximate it (to within a $O(\log k)$ factor) and construct the sets $\mathcal{B}_r$.

We now construct a tree graph, whose nodes are associated with the cover balls: The leaves correspond to singleton balls, hence correspond to the action space. For each leaf $i$ we find an action $a_1(i) \in K$ such that: $i \in B_{2^{-H+1}}(a_1(i)) \in \mathcal{B}_{2^{-H+1}}$. If there is more than one, we arbitrarily choose one, and we connect an edge between $i$ and $B_{2^{-H+1}}(a_1(i))$. We continue in this manner inductively to define $a_r(i)$ for every $a$ and $r < 1$: given $a_{r-1}(i)$ we find an action $a_r(i)$ such that $a_{r-1}(i) \in B_{2^{-H+r}}(a_r(i)) \in \mathcal{B}_{2^{-H+r}}$, and we connect an edge from $B_{2^{-H+r-1}}(a_{r-1}(i))$ and $B_{2^{-H+r}}(a_r(i))$.

We now claim that the metric induced by the tree graph dominates up to factor 4 the original metric. Let $i, j \in K$ such that $\Delta_{\mathcal{T}}(i, j) < 2^{-H+r}$ then by construction there are $i, a_1(i), a_2(i), \ldots a_r(i)$ and $j, a_1(j), a_2(j), \ldots a_r(j)$, such that $a_r(i) = a_r(j)$ and for which it holds that $\Delta(a_s(i), a_{s-1}(i)) \leq 2^{-H+s}$ and similarly $\Delta(a_s(j), a_{s-1}(j)) \leq 2^{-H+s}$ for every $s \leq r$. Denoting $a_0(i) = i$ and $a_0(j) = j$, we have that

$$\Delta(i, j) \leq \sum_{s=1}^{r} \Delta(a_{s-1}(i), a_s(i)) + \sum_{s=1}^{r} \Delta(a_{s-1}(j), a_s(j))$$

$$\leq 2 \sum_{s=1}^{r} 2^{-H+s} \leq 2 \cdot 2^{-H} \cdot 2^{r+1} \leq 4\Delta_{\mathcal{T}}(i, j). \qquad \square$$

## 4.3 Infinite Metric Spaces

Finally, we address infinite spaces by discretizing the space $K$ and reducing to the finite case. Recall that in this case we also assume that the loss functions are Lipschitz.

*Proof of Theorem 9.* Given the definition of the covering dimension $d = \overline{\mathcal{D}}(\Delta) \geq 1$, it is straightforward that for some constant $C > 0$ (that might depend on the metric $\Delta$) it holds that $N_r^{\mathsf{c}}(\Delta) \leq Cr^{-d}$ for

all $r > 0$. Fix some $\epsilon > 0$, and take a minimal $2\epsilon$-covering $K'$ of $K$ of size $|K'| \leq C(2\epsilon)^{-d} \leq C\epsilon^{-d}$. Observe that by restricting the algorithm to pick actions from $K'$, we might lose at most $O(\epsilon T)$ in the regret. Also, since $K'$ is minimal, the distance between any two elements in $K'$ is at least $\epsilon$, thus the covering complexity of the space has

$$C_{\mathrm{c}}(\Delta) = \sup_{r \geq \epsilon} r \cdot N_r^c(\Delta) \leq C \sup_{r \geq \epsilon} r^{-d+1} \leq C\epsilon^{-d+1},$$

as we assume that $d \geq 1$. Hence, by Theorem 7 and the Lipschitz assumption, there exists an algorithm for which

$$\mathrm{Regret}_{\mathrm{MC}}(\ell_{1:T}, \Delta) = \widetilde{O}\left(\max\left\{\epsilon^{-\frac{d-1}{3}}T^{\frac{2}{3}}, \epsilon^{-\frac{d}{2}}T^{\frac{1}{2}}, \epsilon T\right\}\right).$$

A simple computation reveals that $\epsilon = \Theta(T^{-\frac{1}{d+2}})$ optimizes the above bound, and leads to $\widetilde{O}(T^{\frac{d+1}{d+2}})$ movement regret. $\qquad\square$

## Acknowledgements

RL is supported in funds by the Eric and Wendy Schmidt Foundation for strategic innovations. YM is supported in part by a grant from the Israel Science Foundation, a grant from the United States-Israel Binational Science Foundation (BSF), and the Israeli Centers of Research Excellence (I-CORE) program (Center No. 4/11).

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
