[Reviews · NeurIPS 2017]

Reviewer 1



The authors consider the Multi-Armed Bandit (MAB) problem with moving costs, where the underlying metric spaces are quite general. They characterize the regret in terms of a complexity measure of the metric space by showing upper and lower bounds that match within a logarithmic factor. The regret bounds obtained generalize previous known ones. The characterization gives insights into deep understanding of the MAB in a metric space. Very nice results. I wonder the same characterization would apply to the full information setting.

Reviewer 2



This paper studies the non-stochastic multi-arm bandit problem where, in addition to paying the cost of the action chosen, the learner must also pay a cost based on how far he moves from the previous action. Similar settings (with switching costs) have received significant study recently, and the problem is well motivated. This work seems to be the most general and is a generalization of the recent COLT paper [24], which studied metrics defined by Hierarchically well-separated tree. These trees define a metric on a finite space by embedding the elements as leaves in a binary tree and defining distances as the weights on the tree. The show that the algorithm from [24], SMB, obtains regret max{ \sqrt{kT}, C_c^{1/3}T^{2/3}}, where C_c is a covering complexity term the authors defined. By also showing a lower bound of the same order (with a packing complexity C_p instead of a covering complexity), the authors fully characterize the regrets in the finite K case. This is a nice result. The authors also use a connection with covering numbers and Minkowski dimension to derive an upper bound on regret of SMB in the infinite action, Lipschitz loss, case of O(T^(d+1)/(d+2)). This matches a previous lower bound, hence providing a characterization in the infinite case too. These results are interesting and would be a good addition to NIPS. The writing is clear and the paper is well organized, with good discussions adding a lot of insight (e.g. the paragraph from 242). I felt that the last page was rushed, as it really provides the meat of the paper and is quite interesting. I wish the authors could have given more exposition here. Typos: 154: Movement -> movement 258: extra . 261: bottommost -> bottom-most 265: the tildes are really high

Reviewer 3



The authors consider the setting of Multi-Armed Bandits with movement costs. Basically, the set of arms is endowed with a metric and the player pays a price when she changes her action depending on the distance with the latest played action. The main contribution of the paper is to generalize previous work to general metrics. They prove matching (up to log factors) upper and lower bounds for the problem and adapt the Slowly-Moving Bandit Algorithm to general metrics (it was previously designed for intervals only). To my opinion the paper is pretty well written and pleasant to read. The literature is well cited. The analysis seem to be quite similar to the previous analysis of the SMB algorithm. The analysis and the algorithm are based on the same idea of hierarchical trees and the main ingredient of the proof seems to deal with non-binary trees. Though, I think it is a nice contribution. Some remarks: - your notion of complexity only deals with parametric spaces of arms. Do you think you can get results for non-parametric spaces, like Lipschitz functions? - what is the complexity of the algorithm for instance if the space of arms is [0,1]^d? I.e. to build the correct tree and to run the algorithm - your hierarchical tree remind me the chaining technique. Though the later cannot be used for bandits, I am wondering if there is any connexion. - maybe some figures would help to understand the algorithm - is it possible to calibrate eta and the depth of the tree in T? - do you think it may be possible to learn the metric online? Typos: - l171: a dot is missing - l258: "actions K. and" - Algo1: a space is missing; add eta > 0 to the input; - Eq (3): some formating issue - l422: I did not understand the explanation on condition (2)c, don't we have $k > = C$ by definition of C? - 476: tilde are missing

Reviewer 4



This paper considers the non-stochastic Multi-Armed Bandit problem in a setting that the loss contains the cost of switching between actions. It generalizes SMB algorithm of Koren et al. [24] to any metric of actions by using the techniques of covering and packing numbers. The paper is basically the extension of [24] with some new analysis. The theoretical result for the finite metric space recovers the uniform switching cost [2,12] as well as for [24]. Lower bound is also provided, which shows that the theoretical result is tight for any metric space and is a significant contribution. For infinite metric space, the upper bound of regret is also tight. The paper is well written and the logic of the proofs and analysis for generalizing SMB to any metric space is clear and may be viewed as standard. For the infinite metric space, the authors show that the proposed upper bound considering switching costs matches the lower bound that does not assume switching costs, implying that movement costs do not add to the regret of the problem for the infinite space. Maybe the authors can elaborate more about this result, as the proof of the upper bound in this paper is by discretizing the space and reducing to the finite case; yet, the lower bound for finite metrics does consider moving cost. Is something missing here? A high level idea explaining why the result is true would be much appreciated.